# The Impacts of Urbanization and Dietary Knowledge on Seaweed Consumption in China

**DOI:** 10.3390/foods10061373

**Published:** 2021-06-14

**Authors:** Jingsi Peng, Shi Min, Ping Qing, Minda Yang

**Affiliations:** College of Economics and Management, Huazhong Agricultural University, No.1 Shizi Shan Street, Hongshan District, Wuhan 430070, China; pjinsi@webmail.hzau.edu.cn (J.P.); min@mail.hzau.edu.cn (S.M.); qingping@mail.hzau.edu.cn (P.Q.)

**Keywords:** dietary knowledge, seaweed consumption, urbanization, seaweed farming

## Abstract

Edible seaweed, a nutrient-rich and sustainable food, has a long dietary history in China. To get a better understanding of the seaweed consumption of consumers in China, this study investigates the quantity and trend of seaweed consumption of Chinese residents and employs a Tobit model to examine the effects of urbanization and dietary knowledge on seaweed consumption among residents. The results show an increasing trend of household seaweed consumption in China, including both seaweeds consumed at home (SAH) and seaweeds consumed away from home (SAFH). Households in urban areas consumed more seaweeds on average than those in rural areas. Urbanization promotes total household seaweed consumption, including SAH and SAFH, whilst dietary knowledge has a significantly positive impact on total household seaweed consumption and SAH. The findings supplement empirical evidence on the seaweed consumption behavior of Chinese residents and have important policy implications for further promoting Chinese seaweeds consumption in the context of urbanization and increasing dietary knowledge.

## 1. Introduction

Seaweed, as a sustainable marine plant, has great potential to help achieve SDGs (sustainable development goals), such reducing the ecological footprint, eliminating hunger, and helping with food security under the pressure of resource and environment. Seaweed consumption is mainly concentrated in Southeast Asian countries [1,2] and has gradually expanded to Europe and North America in the form of sea vegetables and sea salads [3,4]. Seaweed mainly grows in coastal cities, so there are certain regional differences in consumption [5,6], but the development of urbanization provides market conditions for the settlement of differences. Seaweed is rich in nutrients and biologically active ingredients [7], but few residents have noticed its health effects. Therefore, it is of great significance to explore whether dietary knowledge can improve residents’ seaweed consumption. In addition, since the outbreak of COVID-19, crop production and food security have suffered a huge impact [8,9]. The sustainability of food supply chains has been hit harder, leaving many people without food [10,11,12]. As an important marine plant, seaweed may contribute to improving food security in coastal areas. Hence, given the above potential benefits of seaweed, it will be of positive significance to explore the seaweed consumption of residents.

Edible seaweeds have long been a part of Asian dietary culture, especially in Japan, China, and Korea [1,2]. Notably, China has a long history of food selection and eating methods based on profound knowledge [13], which results in the consumption of diverse seaweeds in China [14]. There are more than 1000 species of seaweed, of which about 50 are available for human consumption [15]. To our knowledge, at present, there are two kinds of edible seaweed food on the market. One is seaweed directly processed as food, such as *Laminaria japonica*, *Undaria pinnatifida*, *Porphyra purpurea,* and *Gelidium amansii* [1]; the other is to use seaweed as raw material to extract its active ingredients or its simple processed products as additives to produce food [16], such as alginate, carrageenan, agar, and so on. It is worth noting that seaweed snacks are also widely popular in the world, and their global sales are expected to increase from USD $1.43 billion in 2020 to USD $2.8 billion in 2028 [17], which promotes the consumption of seaweed in urban areas. Thus, seaweeds are expected to be linked to cooking and food science in the future to promote the health and dietary diversity of residents [18,19].

Seaweed contains a variety of nutrients, such as protein, carbohydrate, minerals, vitamins, and dietary fiber [7], and can provide a large number of nutritious foods for human consumption to alleviate food pressure [20]. In recent years, seaweeds have attracted extensive attention for their splendid bioactive components and properties [21,22]. For example, brown seaweed polysaccharides can regulate the production of short-chain fatty acids and the secretion of mucin by acting on the intestinal flora, thereby enhancing the immune response [23]. Sargassum polycystum can stabilize the blood sugar and blood lipid levels by improving the insulin sensitivity of type 2 diabetic patients, thus playing an antidiabetic role [24]. Many studies on the effects of alginate (a natural polysaccharide carbohydrate extracted from kelp) have shown that it can reduce hunger and energy intake and has a good effect on overweight and obesity [5]. Hence, a diet rich in seaweed is recognized to have potential health benefits [25], and more people are becoming seaweed consumers [26].

As a form of marine culture, seaweed farming seems to be more sustainable than land agriculture. Seaweed farming greatly reduces the use of fresh water, fertilizer, and land [27]; therefore, to some extent, it is environmentally friendly. Seaweed can absorb nitrogen, phosphorus, and carbon dioxide to promote its own growth and produce energy-storage products [28]. Seaweed can also be used to remove inorganic nutrients and mitigate potential adverse environmental impacts [29]. Most seaweed contains chlorophyll, which can use sunlight for photosynthesis and release oxygen for animals to breathe, which is especially important for maintaining the ecological balance of the ocean [30]. Seaweed farming can help the ocean adapt to climate change by damping wave energy and increasing pH to reduce ocean acidification and deoxidation locally [31]. Seaweed farming can be combined with fish, shellfish, and other species to improve the aquaculture environment and thereby increase the output of aquaculture [32], which is conducive to the development of sustainable bio-economy. The value of different species of seaweed is different, but on the whole, seaweed culture has brought considerable environmental benefits and social and economic value. Unfortunately, farmers prefer to grow agricultural products based on economic value [33]. This phenomenon may be associated with consumers’ relatively low demand for seaweeds. Thus, it is necessary to understand how to promote the consumption of seaweeds among residents.

It should be noticed that there are also some different opinions on seaweed farming and seaweed consumption. On the one hand, seaweed farming may affect the surrounding ecosystems, such as predation and competition with wild fish and genes, and disease transmission [4], thus destroying the balance of these ecosystems. On the other hand, the safety of commercial edible seaweed is also worthy of attention. Seaweed decomposes rapidly after absorbing microorganisms [34] and may accumulate harmful compounds, such as heavy metals, minerals, and trace elements [6,35], which may be toxic above a certain limit [36]. A number of studies on the analysis of heavy metal concentrations in edible seaweed products show that seaweed samples contain a large amount of cadmium, nickel, arsenic, and lead, which are beyond the legal provisions [6,37]. This may be due to the ocean environment where seaweed is farmed. Hence, to ensure seaweed safety for coastal developing countries and regions and especially China, which has the highest output and largest area of seaweed culture in the world, it is necessary to formulate new regulations on the supervision of the whole process of seaweed food from culture to marketing [33].

Previous studies have explored seaweed processing techniques [38], seaweed harvest efficiency [39], seaweed nutrition and health benefits [40], the negative effects of seaweed farming [4], and the sustainability of seaweed farming [41]. However, little information is known about seaweed consumption from the perspective of economics, notably in China, the largest consumer in the world. A series of questions are raised. How much seaweed do Chinese residents consume? What is the trend of seaweed consumption of Chinese residents? Is there any heterogeneity in seaweed consumption among Chinese residents? What are the main factors influencing Chinese residents’ consumption of seaweed? This study attempts to answer these questions to improve the understanding of seaweed consumption in China.

Furthermore, this study focuses on two potential determinants of seaweed consumption in China: urbanization and dietary knowledge. China is experiencing rapid urbanization, which has created a rapidly developing market and improved the food accessibility of residents [42,43]. In the past, dietary diversity was closely related to agricultural diversity, especially in rural areas [44], and only coastal residents had relatively easy access to seaweed. However, currently, seaweeds are no longer restricted by region, and even inland residents can buy a variety of seaweed in exquisite packaging from nearby supermarkets at any time. Moreover, the improvement of dietary knowledge is known to help residents adjust their eating behavior and nutrition intake [45,46]. Residents with more education and more dietary knowledge prefer nutritious food when they have sufficient purchasing power [47]. Considering the characteristics of nutrient-rich seaweeds, it is reasonable to hypothesize that dietary knowledge plays an important role in the consumption of seaweeds in China.

The overall goals of this study are to better understand seaweed consumption in China and empirically examine the impacts of urbanization and dietary knowledge on household seaweed consumption. To achieve these objectives, we employed panel data from the China Health and Nutrition Survey (CHNS) for 2004, 2006, and 2009. Based on the established econometric models controlling for the characteristics of household food decision makers and the fixed effects of year and province, this study empirically examined the impacts of urbanization and dietary knowledge on total seaweed consumption (TS), seaweed consumed at home (SAH), and seaweed consumed away from home (SAFH).

The main contributions of this paper include but are not limited to the following aspects. First, according to our literature review, this is the first study on seaweed consumption in China from the perspective of economics, while previous studies mainly focused on issues related to nutrition, disease, and aquaculture. Second, we focus on the effects of urbanization and dietary knowledge on seaweed consumption, which can help improve our understanding of the trend of seaweed consumption in the context of increasing urbanization in China and improving residents’ dietary knowledge. Third, this study not only considers TS but also takes into account SAH and SAFH. As residents’ income and work time increase, an increasing number of people tend to have dinner outside of the home [48]. If we ignore SAFH, the results are likely to be biased.

The rest of this article is organized as follows. Section 2 describes the data sources and the statistics of the variables used in this study. In Section 3, we introduce the Tobit model to estimate the impact of urbanization and dietary knowledge on seaweed consumption. Section 4 reports and discusses the estimation results. The last section summarizes the findings, proposes relevant policy suggestions, and notes the shortcomings of this paper.

## 2. Data and Descriptive Statistics

The data used in this study are from the China Health and Nutrition Survey (CHNS), which were collected by using a face-to-face questionnaire, including samples from nine provinces (Heilongjiang, Liaoning, Jiangsu, Shandong, Henan, Hubei, Hunan, Guangxi, and Guizhou) and three municipalities (Beijing, Shanghai, and Chongqing) in China, conducted in 1989, 1991, 1993, 1997, 2000, 2004, 2006, 2009, 2011, and 2015, respectively. The CHNS provides detailed data on target families and their members as well as their communities, while the modules used in each survey remain as similar as possible. The questionnaire was designed by an interdisciplinary team of social and biomedical scientists with rich theoretical knowledge and practical experience in their respective fields. All the sample data were obtained by field survey, and all the researchers were trained and engaged in nutrition work or participated in other national surveys. Considering the availability of seaweed consumption data, we employed the data from 2004, 2006, and 2009. After dropping missing data, our total sample size is 13,386, and the distribution of samples by province is shown in Table 1. The data on variables used in this study were stored in STATA 16 (StataCorp, College Station, TX, USA), which is also used for descriptive statistics and estimation of empirical models.

We used the food code of seaweed corresponding to the CHNS sample data according to the China Food Consumption Table and then extracted the seaweed consumption data from the CHNS data [49]. The CHNS collected detailed household food consumption data on three random days each week. To facilitate the data presentation, we converted the three days of food records (including food code, dishes, locations, meals, and people) into one month by 3 × 30 and further calculated the per capita consumption of seaweed grams per month within a household.

Figure 1 reports the statistical results of Chinese residents’ seaweed consumption, including TS, SAH, and SAFH, all of which increased from 2004 to 2009. Specifically, TS increased from approximately 59 g/month/person in 2004 to 94 g/month/person in 2009. Notably, SAFH increased over two times from 2004 to 2009, implying the importance of taking into account SAFH in studies on seaweed consumption in China. The remarkable increase in SAFH is probably because seaweed can be used as fast food, such as sushi and seaweed soup, to save time and to fit meals into an increasingly fast-paced life. In Japan and Korea, seaweed is also frequently consumed daily in the traditional diet. The seaweed consumption of Japanese residents has been maintained at 120–180 g/month/person [50], while the intake of seaweeds in Korea was approximately 255 g/month/person [51]. Compared with Japan and Korea, the consumption of seaweed by residents in China is relatively low, indicating that there remains considerable potential for the increase in seaweed consumption in China.

Figure 2 and Figure 3 show seaweed consumption by urbanization. First, the results in Figure 2 indicate the difference in seaweed consumption between rural and urban households. For urban residents, the average SAH is 78.09 g/month/person, and the average SAFH is 27.54 g/month/person, while for rural residents, the average SAH is 48.80 g/month/person, and the average SAFH is 9.27 g/month/person. Obviously, regardless of TS, SAH, or SAFH, the seaweed consumption of urban residents is much higher than rural residents. Second, Figure 3 further shows the statistics of seaweed consumption among the five quantiles of the urbanization development index (UDI), which is computed from the domain population density, economic activity, traditional and modern market availability, transportation, sanitation, communication, housing, and education [52]. Groups 1 to 5 represent low to high levels of UDI. With the increase in UDI, residents consume more seaweed, including TS, SAH, and SAFH. Therefore, seaweed consumption appears to be positively associated with urbanization. This is reasonable because there are relatively good market situations with the development of urbanization, so residents living in urban areas or areas with high UDI more easily access seaweeds through markets than do their counterparts.

The dietary knowledge index (DKI) was calculated by the answers of the respondents to the questions concerning dietary knowledge in the CHNS (Table A1 in the Appendix A). Following previous studies [46,53,54], we calculated the DKI score (1 for a correct answer, −1 for an incorrect answer, and 0 for unknown). The higher the DKI score is, the richer the individual’s dietary knowledge. As shown in Table 2, the level of average dietary knowledge showed an upward trend over time; specifically, it increased sharply from 2004 to 2006 but increased slightly from 2006 to 2009.

To detect the possible correlation between DKI and seaweed consumption, we divided the sample into five equal quantile subsamples according to DKI and calculated the average SAH and SAFH in each subsample. Figure 4 shows the seaweed consumption by DKI. The consumption of seaweeds, regardless of TS, SAH, or SAFH, shows an overall gradually increasing trend. This may be because seaweed is widely recognized as a nutritional and healthy food, and residents with high DKI tend to consume more healthy food [55]. Therefore, it can be hypothesized that dietary knowledge plays an important role in the seaweed consumption of residents [56].

## 3. Methods

To capture the effects of urbanization and dietary knowledge more accurately on seaweed consumption in China, multiple regressions are used to further control for other variables that may also affect household seaweed consumption. Some households have zero consumption of seaweed, so the dependent variable is censored. Following a previous study on food consumption [57,58,59,60], the Tobit model is used to estimate seaweed consumption. The Tobit model is suitable for distributions that are approximately continuous in positive values but contain a portion of result variables with a value of 0 [61]. According to the research content, the left-truncated data model is selected in this paper. Hence, to model the impacts of urbanization and dietary knowledge on seaweed consumption in China, we establish the following equations:(1)$$y_{i}^{*}=\alpha_{0}+\beta_{1}D_{it}+\beta_{2}UDI_{it}+\beta_{3}DKI_{it}+\underset{k}{\sum }\gamma_{ikt}X_{kt}+\delta Z_{it}+\varepsilon_{it}$$
(2)$$y_{i}=\left\{\;\begin{array}{c} y_{i}^{*},\;y_{i}^{*}>0\\ 0,\;y_{i}^{*}\leq 0 \end{array}\right.$$
where $y_{i}$ is the per capita monthly consumption of seaweed with a nonnegative value, denoting SAH, SAFH, or TS; $y_{i}^{*}$ represents the latent variables;$\;D_{it}$ is a dummy variable (1 represents the urban sample, 0 represents the rural sample); $UDI_{it}$ represents the urbanization development index; $DKI_{it}$ represents the dietary knowledge index; $X_{kt}$ represents other socioeconomic and demographic variables that may affect per capita seaweed consumption (including age, gender, education and work of decision-makers of household food consumption, household size, and household income); and $Z_{it}$ denotes the other control variables, including year fixed effects and province fixed effects. To control for price effects, we also take the price of community vegetables (including leafy greens, cabbage, and other vegetables) as a control variable for the price level because of the lack of consumer price variables for seaweed [46], and $\varepsilon_{it}$ is the error term. The means and standard deviations of all these variables among the total sample, urban sample, and rural sample are reported in Table 3. Chi-square tests were also performed to statistically compare sample proportions across urban variables to the rural variables. In the last column, we report our conducted chi-square test, while accordingly the results show that these variables are significantly different between urban and rural.

The estimation strategies are summarized as follows. Based on the established Tobit model, TS is estimated stepwise by adding different control variables. The SAH and SAFH are estimated by the bivariate Tobit model simultaneously, while a likelihood ratio test is accordingly conducted to test the independence between the equations of SAH and SAFH. If it fails to reject the results of the likelihood ratio test of independence, the result means that the error terms of the two equations are unrelated, and the SAH and SAFH should be estimated using the Tobit model separately. Finally, the marginal effects of key independent variables on total seaweed consumption will be calculated.

## 4. Estimation Results

### 4.1. Impacts of Dietary Knowledge and Urbanization on Total Seaweed Consumption

Table 4 presents the Tobit estimation results of total seaweed consumption by stepwise including various control variables. In Model (1), we only include the variables of D, UDI, and DKI. In the second step (Model 2), we control for year and province fixed effects, and, finally, we further add the variables regarding demographic and household characteristics and price control variables (Model 3). The results show that the key variables D, UDI, and DKI always have significant and positive effects on TS regardless of controlling for any other variables. These results indicate that urban residents consume significantly more seaweed than rural residents. With the increase in the urbanization development index, the consumption of seaweeds also increases. Residents with higher dietary knowledge scores consume significantly more seaweed than those with lower dietary knowledge scores. These findings are consistent with the results of descriptive statistics and empirically confirm the hypotheses of this study.

The above findings highlight the important role of urbanization and dietary knowledge in promoting seaweed consumption in China. Similar to the study of [42], urbanization can improve residents’ food accessibility and thereby positively influence the consumption of seaweeds. Moreover, residents with relatively high dietary knowledge normally pay more attention to dietary nutrition intake [46], while seaweed is rich in various nutrients, including dietary fiber, proteins, minerals, and certain vitamins [7]; hence, residents with relatively high dietary knowledge tend to consume more seaweed.

Moreover, several other independent variables, such as age, work status, household size, and household income, also have statistically significant effects on seaweed consumption. As shown in the results of model 3 in Table 4, the age of food-decision makers negatively affects seaweed consumption, revealing that young people appear to consume more seaweed than elderly people. This finding also implies that while seaweed consumption in China shows an increasing trend, population aging in China may hinder the increase, which is similar to the research results that the aging population has a negative impact on China’s meat consumption [62]. In addition, residents living in a family where food-decision makers have jobs will consume more seaweed. This result can probably be explained by the time constraint of food-decision makers [63], i.e., seaweeds are relatively easy to cook, thereby saving time for cooking. Similarly, due to the convenience of cooking seaweeds, households with larger populations tend to consume significantly more seaweeds than those with smaller families. Consistent with the conclusions of the impacts of income on nutrition-rich food in previous studies [64], household income has a significant and positive effect on residents’ seaweed consumption.

### 4.2. Estimations of Seaweed Consumption at Home and Away from Home

A bivariate Tobit model is used to estimate SAH and SAFH simultaneously, while the likelihood ratio test of independence shows that the error terms of the equations of SAH and SAFH are insignificantly correlated (bottom of Table 5). Thus, the equations of SAH and SAFH are independent and should be estimated by Tobit regression separately. Table 5 reports the Tobit regression results of the impacts of D, UDI, and DKI on SAH and SAFH by controlling for vegetable prices at the community level, year fixed effects, and province fixed effects. Consistent with the estimation results for TS, both D and UDI have significant and positive effects on SAH and SAFH. This result indicates that urbanization promotes residents’ seaweed consumption both at home and away from home. However, the impact of DKI on SAH is significantly positive, but the impact of DKI on SAFH is insignificant. This result is reasonable, as the DKI of food-decision makers largely determines the quantity and structure of household food consumption but has no direct impact on residents’ seaweed consumption away from home.

Moreover, the effects of the characteristics of food-decision makers and households on SAH and SAFH are obviously different. Only the variable of education is significantly correlated with SAH, while SAFH is significantly affected by the age and working status of food-decision makers as well as household size and household income. These results indicate that food-decision makers with higher education levels consume more seaweed at home due to their higher concerns about health and nutrition. For seaweed consumption away from home, food-decision makers of older ages consume fewer seaweeds away from home; this is understandable, as they engage in relatively little food consumption away from home [63]. Similar to the results in Table 4, the working status of food-decision makers, household size, and income have significant and positive impacts on SAFH, further indicating that the significantly positive impacts of these three variables on TS occur through their impacts on SAFH.

### 4.3. Marginal Effects of Urbanization and Dietary Knowledge on Seaweed Consumption

Table 6 reports the marginal effects of D, UDI, and DKI on the total seaweed consumption based on the estimation results of model (3) in Table 4. Following the study of Bai et al. [65], the total change of the unconditional expectation of seaweed consumption can be further decomposed into the change of seaweed consumption probability and the change of conditional consumption weighted by seaweed consumption probability. According to the marginal effects of D on TS, urban residents consume seaweeds in greater quantities than rural residents, approximately 48 g/month more on average. The disaggregated results suggest that urban residents have an average of 2.7% higher probability of consuming seaweeds than rural residents, while for residents consuming seaweeds, urban residents consume seaweeds more than approximately 128 g/month on average than rural residents. An increase in UDI of 10 in a community (village) will result in an increase in seaweed consumption of approximately 19 g/month for living residents. Meanwhile, the UDI increase of 10 will cause an approximately 1% increase in the probability of consuming seaweed and an increase of approximately 51 g/month of seaweed consumption. Hence, with the rapid urbanization in China, the demand for seaweed will significantly increase.

While the DKI of household food-decision makers has a significant and positive effect on TS, the marginal effects in Table 6 show that the impact degree is limited. Assuming that all variables are constant at the mean value, a one-point increase in DKI (approximately 24%) will only cause an increase of approximately 3 g/month of seaweed consumption and will increase the probability of consuming seaweeds by 0.2%. For the residents deciding to consume seaweed, a one-point increase in DKI will bring an increase in seaweed consumption of approximately 8 g/month. Considering that the average DKI of residents is relatively high, with the further increase in DKI, the increase in seaweed consumption is limited. Nevertheless, for residents with relatively low DKI, their consumption of seaweeds will continue to increase with the increasing DKI of household food-decision makers.

## 5. Discussion

Considering the potential contributions of seaweed to improving food security and environmental sustainability [7,20,27,32], understanding residents’ seaweed consumption has important implications. Particularly, in China, the huge population and the increasing food demand result in heavy pressure on resource and environment [20,66], while the relative rich ocean resource may provide somewhat alternative food, such as edible seaweed. Unfortunately, the consumption of seaweed by Chinese residents is rarely investigated in previous studies. This study fills the research gap and shows an increasing trend of household seaweed consumption, including both SAH and SAFH. However, the seaweed consumption per capita of Chinese residents remains lower than those of Japanese and Korean residents, implying relatively great improving potentials of seaweed consumption in China.

In addition to controlling for the variables that may affect household food consumption, such as age, education, working time of food-decision makers, and household income [62,63,64], this study pays more attention to studying the impacts of urbanization and dietary knowledge on Chinese residents’ seaweed consumption. As stated by Tian et al. (2019) [42,49], the development of urbanization can increase the availability of food and enrich the food choices of residents; therefore, urbanization may also contribute to the expansion of seaweed consumption. This inference has been empirically confirmed by the results of this study. Moreover, considering that seaweed is a nutrient-rich food [5,7,23,25], the dietary knowledge of food-decision makers also plays a role in the consumption of seaweed. With the economic development in China, the urbanization level and residents’ health awareness will continue increasing [42,43,45,46]. Hence, it can be expected that the consumption of seaweed in China will continue increasing in the context of urbanization and improving dietary knowledge of residents of China. The increasing demand for seaweed may bring new challenges for domestic seaweed farming and have important impact for international seaweed markets.

Finally, all above discussions are in with regard to the positive aspects of seaweed consumption; the food safety issue of seaweed consumption must also be noted [3,4,36,67]. That is, seaweed containing excessive heavy metals due to ocean pollution and unsuitable seaweed farming may severely threaten human health and thereby is inedible [6,37]. Thus, in the context of increasing demand for seaweed, the issues regarding edible seaweed safety regulation require special attention.

## 6. Conclusions

Seaweed, a kind of nutrition-rich food, appears increasingly frequently in people’s diets, while its cultivation is also widely recognized to be more sustainable than land farming. This study investigated residents’ seaweed consumption in China, including seaweed consumption at home and seaweed consumption away from home, particularly focusing on the impacts of urban development and dietary knowledge. The results revealed that, with increasing urbanization, residents in China will consume more seaweed, including seaweed consumption at home and away from home. While improving the dietary knowledge of household food-decision makers can foster seaweed consumption by family members, it does not affect residents’ seaweed consumption away from home.

The findings of this study provide evidence of seaweed consumption in China in the context of urbanization and food-decision makers with improved dietary knowledge. With the advancement of urbanization and the improvement of residents’ dietary knowledge, the consumption of seaweed will continue to increase, which is beneficial to the health of residents. Similarly, given that seaweed consumption is related to human health, it is thus recommended to improve the dietary knowledge of food-decision makers in China, especially for those with relatively low dietary intake. It can be expected that seaweed consumption will be an important alternative food and source of nutrition for residents in China. Moreover, considering the sustainability of seaweed cultivation, it is suggested to pay attention to guaranteeing the natural environment of the sea or ocean, which is the basis of seaweed farming.

Finally, there are several limitations in our study. First, we used CHNS data collected in 2004, 2006, and 2009 for empirical analysis. While the data is quite old and recent data is not available, it provides evidence for exploring the seaweed consumption of Chinese residents in the context of urbanization and increasing dietary knowledge; the CHNS data are relatively concentrated in the eastern and central regions of China, and little information is known about seaweed consumption in areas far from the ocean, such as the western regions in China. Secondly, in China, the consumption of seaweed snacks is increasing, but our research only focuses on edible seaweed as a dish. Future research needs to update data to investigate the recent trend of seaweed consumption in different regions of China and consider the impact of seaweed snacks on seaweed consumption.

## Figures and Tables

**Figure 1 foods-10-01373-f001:**
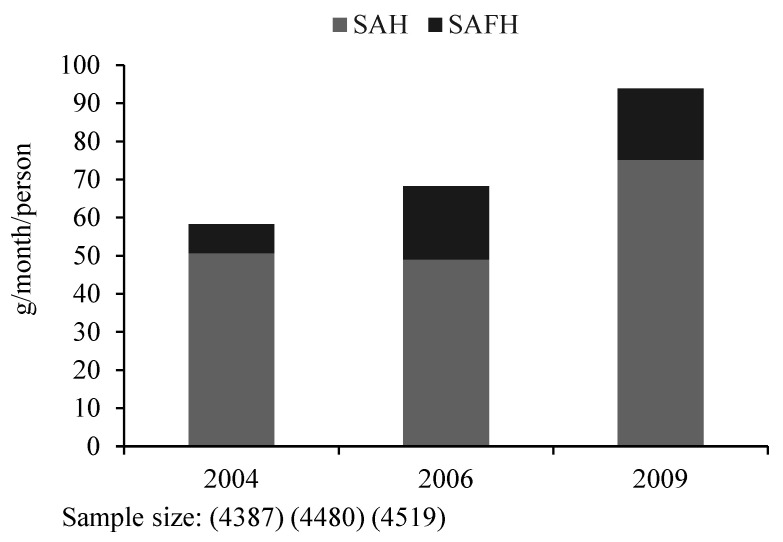
Chinese residents’ seaweed consumption.

**Figure 2 foods-10-01373-f002:**
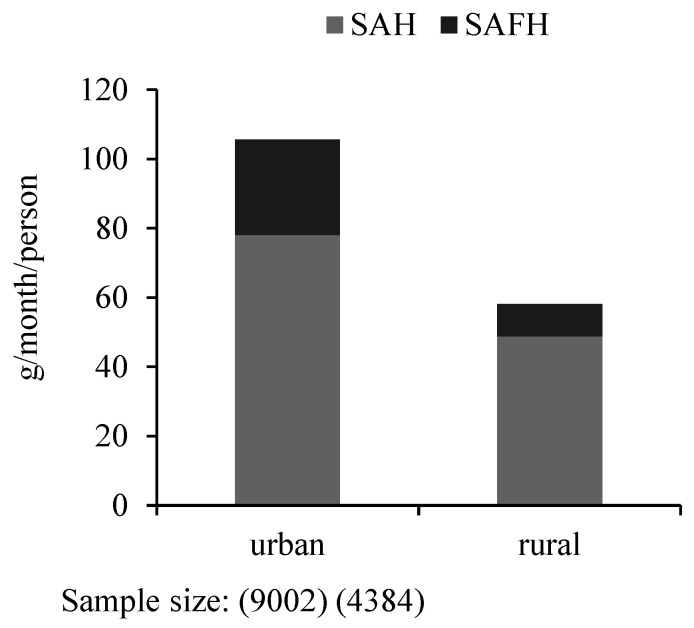
Seaweed consumption between urban and rural residents.

**Figure 3 foods-10-01373-f003:**
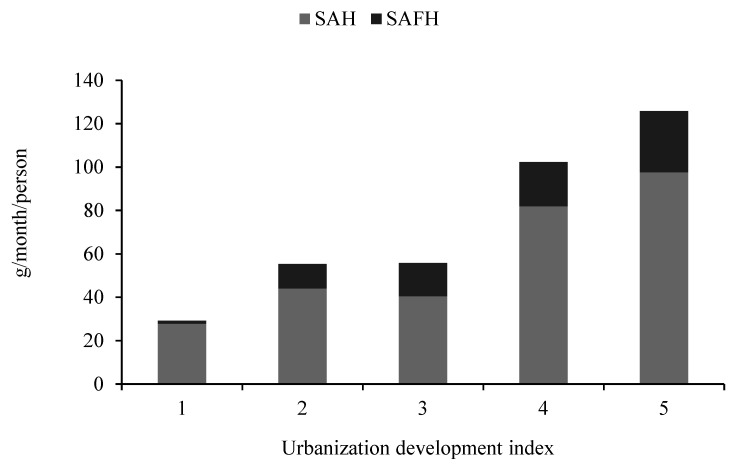
Seaweed consumption by UDI.

**Figure 4 foods-10-01373-f004:**
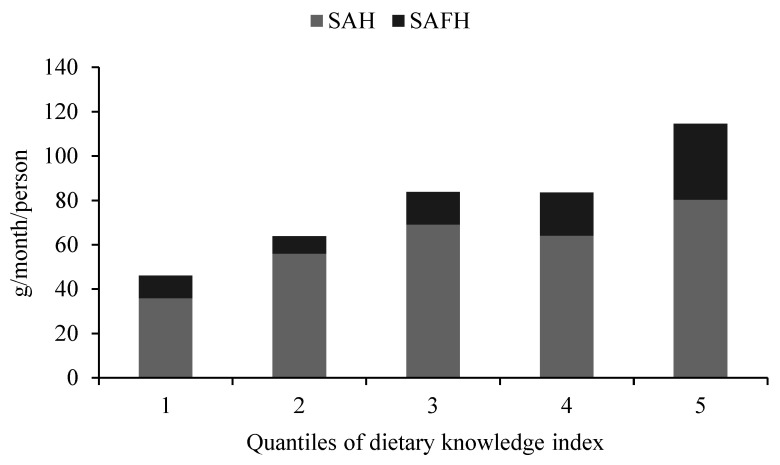
Seaweed consumption by DKI.

**Table 1 foods-10-01373-t001:** Sample size used in this study.

	Sample Size		
Province	2004	2006	2009
Liaoning	482	498	490
Heilongjiang	480	488	488
Jiangsu	490	490	498
Shandong	460	485	480
Henan	496	493	498
Hubei	488	491	500
Hunan	481	507	517
Guangxi	499	502	531
Guizhou	511	526	517
Total	4387	4480	4519

**Table 2 foods-10-01373-t002:** Sample size used in this study.

Year	Obs.	Mean	Std. Dev.	Min	Max
2004	4185	1.58	1.58	−5	9
2006	4240	5.28	2.47	−6	9
2009	4340	5.45	2.54	−5	9

**Table 3 foods-10-01373-t003:** Descriptive statistics of independent variables.

Variables	Description	Total Sample		Urban Sample		Rural Sample		Pearson Chi-Square Test
		Mean	Std. Dev.	Mean	Std. Dev.	Mean	Std. Dev.	
D	1 = urban; 0 = rural	0.3275	0.4695					9.2422 ***
UDI	Urban development index	64.9939	19.6661	78.962	16.0929	58.1914	17.5417	115.5299 ***
DKI	Dietary knowledge index	4.1257	2.8622	4.475	2.8607	3.952	2.8472	88.9574 ***
Age	Age of respondents	48.8962	15.2274	50.7701	14.9825	47.9835	15.263	7.5372 ***
Gender	1 = male; 0 = female	0.2338	0.4233	0.2194	0.4139	0.2408	0.4276	753.3504 ***
Education	Years of schooling	7.6096	4.003	8.701	4.2326	7.0677	3.7685	555.1933 ***
Working status	1 = working; 0 = not working	0.5463	0.4979	0.399	0.4897	0.6199	0.4854	576.8740 ***
Household size	Household members	4.0128	1.7991	3.5297	1.5364	4.2482	1.8693	764.0587 ***
Household income (Yuan in 2015)	Annual income of household	10,290.74	13,427.31	13,592.56	13,717.42	8685.31	12,984.8	61.5514 ***
Vegetable price at community level	Price of vegetables at the community level (Yuan/Jin)	2.4782	1.4268	2.5346	1.4475	2.4507	1.41592	9.2422 ***

Note: ***, *p* < 0.01. 1 Jin = 0.5 kg.

**Table 4 foods-10-01373-t004:** Estimation results on TS.

	Model 1		Model 2		Model 3	
Variables	Coef.	Robust Std. Err.	Coef.	Robust Std. Err.	Coef.	Robust Std. Err.
D	706.42 ***	156.99	879.58 ***	171.94	918.67 ***	175.56
UDI	40.33 ***	4.35	32.12 ***	4.81	37.02 ***	5.21
DKI	122.23 ***	25.96	87.47 ***	33.32	57.29 *	34.05
Age					−27.26 ***	7.08
Gender					−138.06	174.13
Education					22.52	22.79
Working status					391.97 **	171.28
Household size					138.2 ***	43.71
Household income					0.01 **	0.00
Vegetable prices					Controlled	
Year fixed effects			Controlled		Controlled	
Province fixed effects			Controlled		Controlled	
Obs.	12,765		12,765		12,581	
F-statistics	64.00 ***		20.73 ***		12.80 ***	
Pseudo R2	0.0161		0.031		0.0355	

Note: *, *p* < 0.1; **, *p* < 0.05; ***, *p* < 0.01.

**Table 5 foods-10-01373-t005:** Estimation results on SAH and SAFH.

	SAH			SAFH		
Variables	Coef.		Robust Std. Err.	Coef.		Robust Std. Err.
D	557.53	**	254.32	1024.91	***	168.59
UDI	42.28	***	7.60	19.14	***	4.17
DKI	97.49	*	50.45	12.09		28.50
Age	−15.61		10.49	−29.24	***	5.55
Gender (1 = male; 0 = female)	−377.99		267.76	94.78		138.87
Education (years)	62.41	*	33.94	−6.76		17.31
Working status (1 = working; 0 = not working)	173.66		252.20	465.10	***	138.18
Household size	103.53		64.17	110.99	***	35.15
Household income (Yuan in 2015)	0.01		0.01	0.01	***	0.00
Vegetable prices at community level		Controlled			Controlled	
Year fixed effects		Controlled			Controlled	
Province fixed effects		Controlled			Controlled	
Obs.	12,581			12,581		
F-statistics	8.000 ***			5.250 ***		
Pseudo R2	0.0256			0.0771		
Likelihood ratio test of independence	2.326					

Note: *, *p* < 0.1; **, *p* < 0.05; ***, *p* < 0.01.

**Table 6 foods-10-01373-t006:** Marginal effects of key variables on total seaweed consumption.

Variables	Marginal Effects at Observed Censoring Rate		
	Unconditional	Probability	Conditional on
	Expected Value	Uncensored	Being Uncensored
D	47.098	0.027	127.693
UDI	1.898	0.001	5.146
DKI	2.937	0.002	7.964

## Data Availability

The data used in this study can be provided upon request.

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
