# Peer review of "The Impacts of Urbanization and Dietary Knowledge on Seaweed Consumption in China"

_foods, 2021, doi:10.3390/foods10061373_

Round 1

Reviewer 1 Report

My comments are the following:

  1. The aim is not clearly defined in the abstract.
  2. The abstract is missing the exact findings of the study.
  3. How interviews were collected, in person or online?
  4. Which software was used for the statistical analysis?
  5. Why chi-square test was not used.
  6. The part of the study should consider the health hazardous side of the seed weed consumption. The following reference should be used: Kulawik, P., Dordevic, D., GambuÅ›, F., Szczurowska, K., & ZajÄ…c, M. (2018). Heavy metal contamination, microbiological spoilage and biogenic amine content in sushi available on the Polish market. Journal of the Science of Food and Agriculture, 98(7), 2809-2815.

Author Response

Dear reviewer: We sincerely thank you for taking the time to review our paper and for putting forward many constructive comments and suggestions. We believe that these comments and suggestions not only helped us improve the manuscript, but also provided insights for future research. Below we outline our attempts to address your comments. We read the comments carefully one by one and updated the manuscript in response. We used MS Word's "Track Changes" tool for your convenience, so you can see the revisions in the document. The specific line numbers have also been included for the modifications.

Reviewer 2 Report

Peer-review on the manuscript: “The Impacts of Urbanization and Dietary Knowledge on Sea-weed Consumption in China” – Jingsi Peng, Shi Min, Ping Qing and Minda Yang. – Foods, 2021, 10,

This paper discusses the impacts of urbanization and dietary knowledge on seaweed consumption in in the eastern and central regions of China. Panel data from the China Health and Nutrition Survey (CHNS) was used. Edible seaweed is a nutrient-rich and sustainable food and has a long history of consumption in the coastal regions in Asia, especially in Japan, China, and Korea. The use of cultivated seaweeds as food deserves increasing interest in other parts of the World, too. Extracts from seaweed can be used in a wide range of product applications, such as animal feed, toothpaste, cosmetics and medicines. This paper provides an exiting example to the importance of dietary knowledge and to the not always negative effects of urbanization.

Major comments.

At least one sentence about the food safety of commercialized edible seaweeds would be useful in the “1. Introduction”. Also, the Chinese and international regulatory framework, the Best Management Practices for farming and processing and the use of seaweed as human food should be explained (Grebe et al., 2019). How the organic certification works for fresh and processed seaweed, e. g. Soil Association Organic, Bristol, England? Seaweeds are characterized by a rapid microbial decomposition once harvested (Enríquez et al., 1993). They may accumulate undesired compounds such as heavy metals, minerals and trace elements which may be toxic above a certain limit, e. g. iodine. (Almela et al. 2006; Besada et al. 2009; Lüning – Mortensen, 2015). Seaweed species, as well as coastal water qualities, vary by regions. Brown seaweeds can contain a high level of iodine while green and red seaweeds, including the popular sushi seaweed nori, are relatively low in iodine (Zheng et al., 2019).

Also, the structure of the consumed seaweed terminology needs to be explained. Alginate consumption is mentioned on the bottom of the first page. But, for some readers it would not be clear how they relate to seaweeds. Seaweed is a term which can be used to describe many different marine-based species of plants and algae, for example kelp, but not Spirulina. For details, see Lobban - Harrison, 1994): Morphology, life histories, and morphogenesis.

There are also differences in environmental values among seaweed species, but farmers like produce according to the economic value (Zheng et al., 2019).

Questions

  1. What is the possibility for integration of seaweed farming with other farming technologies, e. g. fish farming?
  2. What is the price of seaweed in China, compared to its competitor substitute products? It is mentioned in the article that inland residents can buy a variety of seaweed in exquisite packaging from nearby supermarkets at any time. This explains that, seaweed is not a niche product in the central and eastern regions in China. In Europe, these products are mainly sold in organic or health food shops and the price varies greatly with product category (Le Bras et al., 2015). Seaweed is occasional used as a traditional ingredient in European coastal areas (Bouga – Combet, 2015).

It is quoted (Ren et al., 2019: Family Income and Nutrition-Related Health: Evidence from Food Consumption in China.) in the paper, that consumer price variables for seaweed is lacking. How can it be, if seaweeds are sold in supermarkets? Seven retailers were considered to be large supermarkets with online presence, selling 30% of the seaweed product range in an earlier study in the UK (Bouga - Combet, 2015). Anyway, the use of vegetable prices looks reasonable. The categorization of potato as a vegetgable in this case is always a question.

  1. Seaweed snacks are also available (Fior Markets, 2021), but it cannot be compared to vegetables. Do these kinds of snacks play a role in the popularity of seaweeds in urban area?

References

Almela, C., Clemente, M. J., Velez, D. et al (2006): Total arsenic, inorganic arsenic, lead and cadmium contents in edible seaweed sold in Spain. Food and Chemical Toxicology, 44, 11, 1901–1908

Besada, V., Andrade, J. M., Schultze, F. et al (2009) Heavy metals in edible seaweeds commercialised for human consumption. Journal of Marine Systems, 75, 1–2, 305-313.

Bouga, M., Combet, E. (2015): Emergence of Seaweed and Seaweed-Containing Foods in the UK: Focus on Labeling, Iodine Content, Toxicity and Nutrition. Foods, 4, 2, 240–253.

Engle, C., Kotowicz, D., McCann, J., Cygler, A. (2018): Potential Supply Chains for Seaweed Produced for Food in the Northeastern United States. Final Report, USDA FSMIP Award No. 16FSMIPR10004,

Enríquez, S., Duarte, C. M., Sand-Jensen, K. (1993): Patterns in decomposition rates among photosynthetic organisms: the importance of detritus C:N:P content. Oecologia, 94, 4, 457–471.

Fior Markets (2021): Seaweed Snacks Market by Type (Flakes, Nori Sheets, Chips, Bars, Others), Source (Red, Green, Brown), Distribution Channel (Specialty Stores, Hypermarkets & Supermarkets, Online, Convenience Stores), Region, Global Industry Analysis, Market Size, Share, Growth, Trends, and Forecast 2021 to 2028, Fior Markets, Maharashtra, India, https://www.fiormarkets.com/reportdetail/419172/request-sample

Grebe, G. S., Byron, C. J., St. Gelais, A., Kotowicz, D. M., Olson, T. K. (2019): An ecosystem approach to kelp aquaculture in the Americas and Europe. Aquaculture Reports, 15, 100215.

Le Bras, Q., Lesueur, M., Lucas S et al. (2015): Etude du marché français des algues alimentaires. Panorama de la distribution en magasins. Programme IDEALG Phase 2: Les publications du Pôle halieutique, AGROCAMPUS OUEST, 36, 42.

Lobban, C., Harrison, P. (1994): Morphology, life histories, and morphogenesis. In Seaweed Ecology and Physiology (pp. 1-68). Cambridge University Press, Cambridge, UK

Lüning, K., Mortensen, L. M, (2015): European aquaculture of sugar kelp (Saccharina latissima) for food industries: iodine content and epiphytic animals as major problems. Botanica Marina, 58, 6, 449-455.

Stévant, P., Rebours, C., Chapman, A. (2017): Seaweed aquaculture in Norway: recent industrial developments and future perspectives. Aquaculture International, 25, 1373-1390.

Zheng, Y., Jin, R., Zhang, X., Wang, Q., Wu, J. (2019). The considerable environmental benefits of seaweed aquaculture in China. Stochastic Environmental Research and Risk Assessment. 33, 1203–1221.

Author Response

(The authors gave the same response as above.)

Reviewer 3 Report

MS 1237887: The Impacts of Urbanization and Dietary Knowledge on Seaweed Consumption in China.

The ms needs some improvements, below are my comments:

Authors should make their MS a universal and link this with the local issue. Do not localize your investigation, if you want to publish it in a high impact factor journal such Foods.

Add one more strong keyword

Authors must add some recent published articles 2021 and 2020, because only one citation was used for 2020, ref 38.

Authors should write a paragraph in the introduction about the food security and seaweed as well as link this with the impact of COVID 19 on food security. They should also read and cite this article: https://doi.org/10.14393/BJ-v36n4a2020-54560

Why authors employed panel data from the China Health and Nutrition Survey (CHNS) for 2004, 2006 and 2009? Why from these three specific years? Why not from 2019,2020 and 2021? Particularly, author can link this with covid 19.

The authors wrote too much text about the aims of the study at the end of the introduction, they should summarize this long text.

What is the different between years in L82-83 and L110? Years are not same!

Figures are not well presented, authors should see how other authors make a good and high quality figures. Just, check articles published in MDPI journals.

Where is the discussion section?

In conclusion, why authors use citations?

Author Response

(The authors gave the same response as above.)

Round 2

Reviewer 1 Report

The article is improved.

Reviewer 3 Report

The MS has been improved, thanks for the reviewers and good luck for them.